

# Entropy-driven phase behavior of associative polymer networks

Lorenzo Rovigatti[1,2] and Francesco Sciortino[1]

**1** Department of Physics, Sapienza Università di Roma,
Piazzale A. Moro 2, IT-00185 Roma, Italy
**2** CNR-ISC Uos Sapienza, Piazzale A. Moro 2, IT-00185 Roma, Italy

## Abstract

Polymer chains decorated with a fraction of monomers capable of forming reversible bonds form transient polymer networks that are important in soft and biological systems. If chains are flexible and the attractive monomers are all of the same species, the network formation occurs continuously as density increases. By contrast, it has been recently shown [L. Rovigatti and F. Sciortino, Phys. Rev. Lett. 129, 047801 (2022)] that, if the attractive monomers are of two different and alternating types, the entropic gain of swapping intra-molecular bonds for inter-molecular connections induces a first order phase transition in the fully-bonded (*i.e.* low-temperature or, equivalently, large monomer-monomer attraction strength) limit and the network forms abruptly on increasing density. Here we use simulations to show that this phenomenon is robust with respect to thermal fluctuations, disorder and change in the polymer architecture, demonstrating its generality and likely relevance for the wide class of materials that can be modelled as associative (transient) polymer networks.

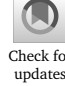

## Contents

Transient polymer networks (TPNs), where polymeric macromolecules bind to each other through reversible bonds, are interesting as synthetic materials, as well as models for many biological systems, ranging from the actin cytoskeleton [1] to membraneless organelles [2,3]. The bonding mechanism that drives the formation of the network can have either a non-covalent origin, *e.g.* metal complexation [4], ionic or hydrophobic interactions [5], hydrogen bonding [6,7], or even a covalent nature, provided the system is endowed with a bond-exchange mechanism that makes it possible to rearrange the network while retaining the same number of bonds [8,9].

From a modelling perspective, even the simplest description of transient polymer networks should retain at least two main ingredients: (i) a polymeric nature, and (ii) a bonding mechanism that drives the formation of transient inter- and intra-molecular connections. The interplay between these two attributes, where the tendency to self-assemble is modulated by the internal flexibility of the polymers, dramatically affects the static and dynamic behaviour of the resulting material compared to non-associating polymeric systems.

Here we study a simple model of monomer-resolved transient polymer networks, where polymeric chains have a sticker-spacer-like architecture [10,11]: reactive monomers (the stickers) are separated by inert, *i.e.* purely-repulsive, monomers (the spacers). Here, as in other sticker-spacer models [12,13], stickers cannot be involved in more than one bond each. This is a condition that is fulfilled by many naturally-occurring and synthetic systems, where it can be enforced by, *e.g.*, lock-and-key biomimicking mechanisms [14], or metal complexation [4,15]. In the diluted regime the polymers rarely interact with each other, and the great majority of the bonds are formed between reactive monomers belonging to the same chain (intra-molecular bonds). If the bonding strength is sufficiently large, all the reactive monomers are engaged in a bond and the system is in a *fully-bonded* state composed by isolated polymers, akin to single-chain nanoparticles (SCNPs) [16]. If the reactive monomers are all of the same type, in the scaling limit polymer theories predict the absence of phase separation [10]: as the polymer concentration increases, polymers start to interact with each other, and intra-molecular bonds are swapped for inter-molecular bonds, forming a (fully-bonded) transient polymer network without encountering any phase separation. Experimental [17,18] and numerical [19–21] evidence supports these theoretical predictions.

If polymers are flexible, in a system with a fixed number of polymers the energy is proportional to the number of bonds, and this number at large bonding strength is independent of concentration, since all possible bonds are always formed. The thermodynamics of the system is thus fully determined by the balance between different entropic contributions that control the ratio of intra- to inter-molecular bonds [10]. We recently showed numerically that the balance can be modulated to induce phase separation by introducing reactive monomers of different types that, if placed appropriately along the polymers, entropically penalise intra-molecular bonds in favour of inter-molecular bonds [21]. When bonds can be formed only between reactive monomers of the same type, decorating the polymers with alternating, equally-spaced reactive sites is enough to drive a phase separation that has a fully-entropic origin [21]. A combinatorial entropy-driven attraction has also been found to play an important role in gold-nanoparticles grafted with self-complementary DNA strands [22] as well as in other soft-matter systems [23] (for a recent review see for example [24]).

In this article we examine in more details this class of polymeric systems to show that this entropy-driven gas-liquid phase separation is a very robust phenomenon, which is not only found in the fully bonded case, but also when the number of bonds in the system is thermally modulated (*i.e.* when the bonding strength is comparable to the thermal energy). We also explore the case in which the arrangement of the reactive monomers is disordered, confirming the presence of the phase separation. Finally, we show that even in systems composed by polymer rings rather than chains, the same phenomenon is observed.

# 1  Methods

We simulate systems composed by $N_c = 100$ polymers made of a fixed number $N_m$ of monomers, of which $N_r$ are reactive. Each reactive monomer bears a type that dictates the way it interacts with other reactive monomers. Focusing only on the sequence of the $N_r$ reactive monomers, we refer to systems made of polymers containing $N_r$ monomers of the same type

as $A_{N_r}$ (*e.g.* $A_{24}$), and to systems made of polymers containing $N_r$ monomers of alternating types as $(AB)_{N_r/2}$ (*e.g.* $(AB)_{12}$).

We study linear polymers (*i.e.* chains) made of $N_m = 243$ monomers, with both ordered and disordered arrangements of the reactive monomers. The ordered chains start and end with 6 inert monomers, and then the $N_r$ reactive monomers are placed equispatially, with 9 inert monomers separating each pair of neighbouring reactive monomers. Note that we choose a slightly different system than the one in Ref. [21] to show that small differences in the number of inert monomers separating neighbouring reactive monomers (9 *vs.* 10) and the presence of short inert segments at the beginning and at the end of the chain, absent in Ref. [21], do not alter the phenomenon we observe. In the case of disordered arrangements, the position of the reactive monomers are generated as follows: (i) a non-reactive monomer $i$ is selected at random; (ii) if $i$ is separated by more than $S$ inert monomers from any other reactive monomer then $i$ is flagged as a reactive monomer; (iii) if the number of reactive monomers is equal to the target $N_r$ value the procedure is stopped, otherwise steps (i)-(iii) are repeated until such a condition is fulfilled. By contrast, polymer rings are made by $N_m = 264$ monomers and $N_r = 24$ reactive monomers, with pairs of neighbouring reactive monomers being separated by 10 inert monomers.

Consecutive monomers along the chain are considered as bonded neighbours and interact through the classic Kremer-Grest potential [25], in which the connectivity is provided by a finitely extensible nonlinear elastic (FENE) term:

$$V_{\text{FENE}}(r) = -\frac{1}{2}K d_0^2 \ln\left(1 - \frac{d_0}{r}\right)^2 , \tag{1}$$

where $r$ is the distance between the two monomers, $d_0 = 1.5\,\sigma$, $K = 30\,\epsilon_{LJ}/\sigma^2$ and $\epsilon_{LJ}$ and $\sigma$ are the units of energy and length, respectively. The FENE term is complemented by an excluded volume contribution modelled through the WCA potential [26]:

$$V_{\text{WCA}}(r) = \begin{cases} 4\epsilon_{LJ}\left[\left(\frac{r}{\sigma}\right)^{12} - \left(\frac{r}{\sigma}\right)^{6} + \frac{1}{4}\right], & r < 2^{1/6}, \\ 0, & \text{otherwise.} \end{cases} \tag{2}$$

The interaction between pairs of monomers that are not bonded neighbours has always a repulsive contribution given by Eq. (2). However, if the two monomers are (i) reactive and (ii) of the same type, their interaction also features an attractive part whose functional form is the one originally proposed by Stillinger and Weber [27] for their model for silicon and often used to model short-range attraction in soft-matter systems [28, 29]. It reads

$$V_{\text{bind}}(r) = \begin{cases} C\epsilon\left[D\left(\frac{\sigma_s}{r}\right)^4 - 1\right]e^{\sigma_s/(r-r_c)}, & r < r_c, \\ 0, & \text{otherwise,} \end{cases} \tag{3}$$

where $\sigma_s = 1.05\,\sigma$, $r_c = 1.68\,\sigma$, $C = 8.97$, $D = 0.41$ and the strength of the attraction (the depth of the minimum) is $\epsilon$. In the following we will set $\beta\epsilon = 20$ as in Ref. [21] to obtain systems that are essentially fully bonded, or smaller values to probe thermal effects.

The single-bond-per-reactive-monomer condition is enforced by an additional three-body term that acts against the formation of triplets formed by bonded monomers, $V_{3b}$ [30]. Specifically, $V_{3b}$ provides a repulsive contribution that tends to match the energy gain associated to the formation of a second bond. The net effect is that a monomer $i$ approaching a pair of bonded particles $j$ and $k$ moves along an almost flat energy hypersurface, so that final configurations in which $i$ is bonded either to $j$ or $k$ while the third particle is free to leave are not separated from the initial configuration from any potential energy barriers.

The specific form of the $V_{3b}$ contribution is

$$V_{3b}(r_{ij}, r_{ik}) = \epsilon \sum_{ijk} V_3(r_{ij}) V_3(r_{ik}), \tag{4}$$

where $r_{ij}$ is the distance between particle $i$ and $j$, and the sum runs over all bonded triplets, defined as groups of particles where $i$ is closer than $r_c$ to both $j$ and $k$. The two-body potential $V_3(r)$ is defined in terms of $V_{\text{bind}}(r)$ as follows:

$$V_3(r) = \begin{cases} 1, & r \leq \sigma_s, \\ -\frac{V_{\text{bind}}(r)}{\epsilon}, & \text{otherwise,} \end{cases} \tag{5}$$

where $\sigma_s$ is the position of the minimum of $V_{\text{bind}}(r)$.

In the following we calculate equations of state by means of constant-temperature molecular dynamics simulations of $N_c = 100$ polymers where we fix the density and compute the virial pressure over the course of long simulations ($4 \times 10^8 - 4 \times 10^9$ time steps, depending on the specific system). We also run direct-coexistence simulations to estimate the density of coexisting phases where we first equilibrate a system at some density at which it does not phase-separate, then we double it along the $x$ direction and then increase the size of the simulation box along $x$ by a factor of 6 and finally we run constant-volume simulations and let the system equilibrate. We then compute the average density profile and use it to estimate the density of the coexisting gas and liquid phases.

In the simulations we vary the attraction strength of the reactive sites $\epsilon$ (see Eq. 3). We fix the temperature so that $\beta \epsilon_{LJ} = 1$, where $\beta$ is the inverse of the Boltzmann constant times the temperature, by coupling the system to an Andersen-like thermostat [31]. The integration time step is set to $\Delta t = 0.003$ in units of time, which are given by $\sigma \sqrt{m/\epsilon_{LJ}}$, where $m$ is the mass of a monomer.

## 2 Results

We start by considering the effect that the polymer architecture has on the average loop length, *i.e.* the average chemical distance between two reactive monomers involved in an intramolecular bond. Table 1 shows this quantity for ordered and (selected) disordered chains, as well as for rings, for the highest probed density ($\rho \sigma^3 = 0.00049$) and strongest attraction

Table 1: Average intra-molecular loop length for different systems and its associated standard deviation computed at $\rho \sigma^3 = 0.00049$ and $\beta \epsilon = 20$. The top six rows pertain to chains, whereas the bottom two refer to ring polymers.

| System | Average loop length | Standard deviation |
|---|---|---|
| $A_{24}$ | 20.9 | 2.3 |
| disordered $A_{24}$, $S = 4$ | 15.2 | 0.9 |
| disordered $A_{24}$, $S = 1$ | 11.4 | 0.7 |
| $(AB)_{12}$ | 38.0 | 3.1 |
| disordered $(AB)_{12}$, $S = 4$ | 33.5 | 1.2 |
| disordered $(AB)_{12}$, $S = 1$ | 24.2 | 1.0 |
| $A_{24}$ rings | 31.4 | 2.1 |
| $(AB)_{12}$ rings | 46.8 | 3.1 |

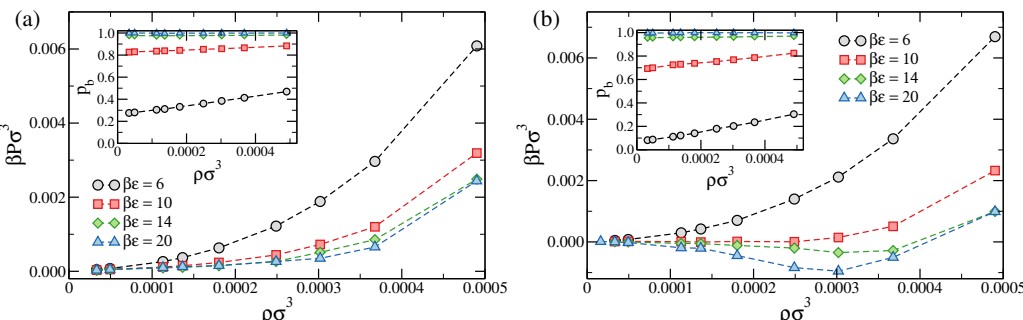

Figure 1: The equations of state of systems for the (a) $A_{24}$ and (b) $(AB)_{12}$ polymers for different values of the bonding strength $\beta\epsilon$ as a function of the polymer density $\rho$. The insets show the fraction of formed bonds $p_b$ for the same simulations.

strength ($\beta\epsilon = 20$). As we will discuss and show below, under these conditions all the systems are homogeneous (no signs of phase separation) and essentially fully bonded, so that the average loop length can be considered a proxy for the entropic cost of forming an intramolecular bond. Focussing on the chain systems (top six rows), it is clear that ($AB$) chains tend to form larger loops compared to $A$ chains, which causes an increased polymer-polymer effective attraction that can drive phase separation [21]. However, in the presence of disordered arrangements of the reactive monomers the average loop length decreases. In the case of maximum disorder ($S = 1$), the average loop length of the $(AB)_{12}$ system is comparable to the value found for the ordered $A_{24}$ chains and, in fact, the consequential reduced cost of intra-molecular bonds is enough to suppress phase separation (see below).

Table 1 also shows results for ring systems, for which going from one to two types of reactive monomers has the same qualitative effect observed for chains: the entropic cost of forming an intra-molecular bond goes up, and therefore the average loop length increases, causing a larger ring-ring effective attraction (see below for the consequences on the thermodynamics). However, the values in this case are larger than what observed in chains, owing to the smaller radius of gyration of (and therefore to the increased probability of intra-molecular contacts in) rings compared to chains of the same chemical size [32].

We start by evaluating the effect that the strength of the attraction $\beta\epsilon$ has on the equations of state of the $A_{24}$ and $(AB)_{12}$ systems. Defining $p_b$ as the fraction of formed bonds, we see in the insets of Figure 1 that at the highest probed $\beta\epsilon$, $p_b \approx 1$ and therefore all possible bonds are formed. Under these conditions, for which we find results compatible with those of Ref. [21], the effective attraction between the chains is the strongest for both classes of systems, as demonstrated by the lowest pressures experienced by the system (see the equations of state shown in the main panels of Figure 1). As $\beta\epsilon$ decreases, the fraction of formed bonds becomes smaller and $p_b$ acquires a stronger dependence on the polymer density $\rho$, and for a given value of the density the associated pressure becomes larger, signalling that the effective chain-chain interaction becomes progressively more repulsive. Nevertheless, the $(AB)_{12}$ system exhibits an equation of state that has, in a wide range of $\beta\epsilon$ a non-monotonic dependence on $\rho$. The non-monotonic dependence of the pressure with density is suggestive of a phase separation, from very large values of $\beta\epsilon$ (where $p_b = 1$) down to values of $\beta\epsilon \approx 10$ (for which $p_b \gtrsim 0.7$).

We confirm that the systems displaying a non-monotonic equation of state do exhibit phase separation by using direct-coexistence simulations. Figure 2 shows the densities of the two coexisting phases as extracted by direct coexistence, as well as representative snapshots of configurations obtained for selected values of $\beta\epsilon$. For $\beta\epsilon = 10$, the interface between low and high density regions becomes very broad, and it disappears for $\beta\epsilon = 8$, suggesting that for $\beta\epsilon \approx 10$ the system is close to the critical point. Thus, for a wide range of attraction

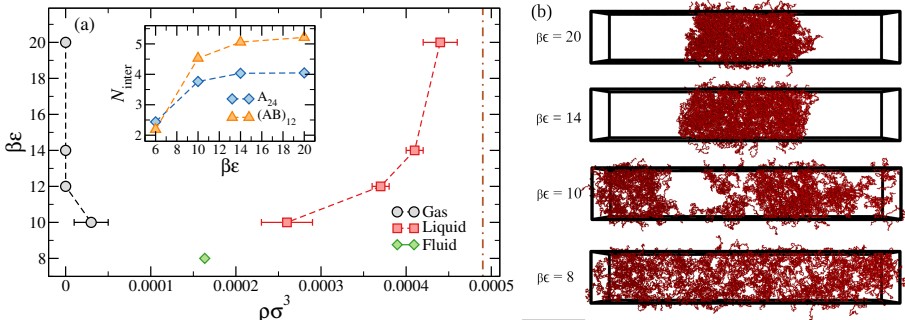

Figure 2: (a) The density of the coexisting phases of the $(AB)_{12}$ system for different values of $\beta\epsilon$, estimated with direct coexistence simulations. The green diamond marks a system where no phase separation was observed. Inset: the number of inter-molecular bonds per chain for the $A_{24}$ and $(AB)_{12}$ systems at the density marked by the dash-dotted vertical line of the main panel ($\rho\sigma^3 = 0.00049$), as a function of $\beta\epsilon$. (b) Snapshots of $(AB)_{12}$ systems for selected values of $\beta\epsilon$.

strengths, from the fully bonded case to the $p_b \approx 0.7$ limit, a clear phase-separation is observed, confirming that the change from the low-density isolated chains to the percolating network phase takes place via a first-order transition only in the case in which reactive monomers of different type alternate along the chain, demonstrating the robustness of the mechanism [21] that drives the phase transition with respect to thermal fluctuations.

In order to gain more insight on the differences between the $A_{24}$ and $(AB)_{12}$ systems, we have also computed the number of inter-molecular bonds formed by each chain, $N_{\text{inter}}$, for the same two systems of Figure 1. The inset of Figure 2(a) shows $N_{\text{inter}}$ for the $A_{24}$ and $(AB)_{12}$ systems at $\rho\sigma^3 = 0.00049$ (the highest investigated density), as a function of $\beta epsilon$. At this high density no signs of phase separation have been detected in either systems. At the lowest value of $\beta\epsilon$ $N_{\text{inter}}$ is comparable between the two systems, although in the $A_{24}$ more inter-molecular bonds are present in virtue of the higher bonding probability (see insets of Fig. 1). However, as the attraction reaches $\beta\epsilon = 10$, the number of inter-molecular bonds per chain in the $(AB)_{12}$ system, for which direct coexistence results demonstrate phase separation at lower density, experiences a steep increases that is not present in the $A_{24}$ system and that drives the transition. We also find that the $(AB)_{12}$ system with $\beta\epsilon = 10$ and $\rho\sigma^3 = 0.00025$, which, according to direct coexistence results, is near-critical and close to the density of the liquid phase, has a $N_{\text{inter}} \approx 3$, making it a rather low-valence and sparsely-connected system.

Up to now, as well as in Ref. [21], the number of reactive sites on each chain was chosen even to guarantee that a fully bonded configuration could be reached at large values of $\beta\epsilon$ even in a dilute phase of isolated polymers. Similarly, reactive sites were equally spaced along the chain, creating an ordered arrangement. We now consider systems composed by chains that have either an odd number of attractive monomers or whose attractive monomers are arranged in a non-ordered fashion. For this comparison we limit ourselves to the large bonding strength limit, by setting $\beta\epsilon = 20$. To consider chains having an odd number of attractive monomers we simulate two types of systems: the $A_{23}$ system and a mixture of $(AB)_{11}A$ and $(BA)_{11}B$ chains to retain the same total number of $A$ and $B$ monomers. If $N_r$ is not even then a single chain cannot fold on just itself to satisfy all its bonds, but requires another chain, making the fully-bonded gas phase composed by two-chain dimers rather than single chains. This effect is apparent in the equation of state at low density (see inset of Figure 3(a)), which shows that the $N_r = 23$ system has, at low density, a lower pressure compared to the ordered $N_r = 24$ system.

The equation of state is shown in the panels (a) and (b) of Figure 3. Fig. 3(a) refers to the case of same-type monomers, where a monotonic pressure-density relation is observed. While

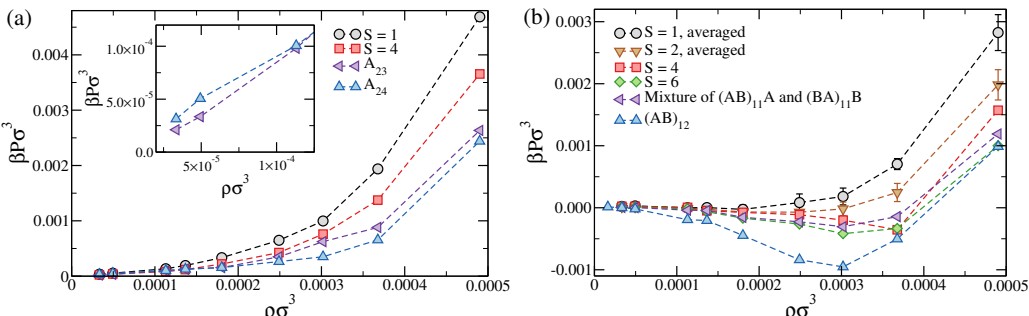

Figure 3: The equations of state of systems made of polymers having ordered, disordered ($S < 9$) and "defective" ($N_r = 23$) arrangements of attractive monomers as a function of $\rho$. Panel (a) refers to $A_{23}$ and $A_{24}$ polymers, with the inset showing the low-density behaviour of the defective and ordered arrangements; panel (b) refers to $(AB)_{12}$ polymers. Note that in this case the $S = 1$ and $S = 2$ curves are obtained by averaging over three different realisations, with the error bars indicating the standard deviation. In all simulations we fix $\beta\epsilon = 20$.

at low density the $A_{23}$ system has a pressure lower than the $A_{24}$ due to the dimer-formation process, at large density the opposite behavior is observed. This suggests that the reduced number of available associative monomers results in a larger effective repulsion between the polymers.

Fig. 3(b) shows that, as previously found for the $(AB)_{24}$, in the case of two different reactive sites the equation of state is non-monotonic. However, the driving force for phase-separation (the extent of the tensile strength) decreases on considering an odd number of reactive sites, which again is due to the smaller number of available associative monomers.

Finally, we also consider the effect of non-ordered arrangements of the attractive monomers for both the $A_{24}$ and $(AB)_{12}$ systems. The equations of state obtained for different values of $S$ are shown in Figure 3. Since $S$ controls the smallest distance between attractive monomers, decreasing $S$ also decreases the entropic penalty of forming intra-molecular bonds. As a result, the driving force to swap intra- with inter-polymer bonds is smaller and the overall inter-polymer connectivity goes down, and systems become more repulsive.

This is, on average, also true for the $(AB)_{12}$ systems. However, especially for small values of $S$, there is a quite strong dependence on the realisation of the disorder, *i.e.* on the specific topology. Here we evaluate the equations of state for three $S = 1$ and three $S = 2$ systems and plot the average equations of state in Figure 3(b), where the error bars are the associated standard deviations. We see that, on average, the pressure at a given density decreases as $S$ increases. For all the cases we studied, $(AB)_{12}$ systems with $S > 1$ always exhibit non-monotonic equations of state, demonstrating that the tendency to phase separate is quite a robust phenomenon, and vanishes only if the attractive monomers are very close along the chain.

Since the architecture of the polymeric objects has an important role in dictating their collective behaviour, we also simulate systems made of rings decorated with $N_r = 24$ reactive monomers arranged in an ordered manner. Recent numerical results have shown that unentangled ring polymers decorated with a single type of reactive monomers behave like chains, in that they do not experience a gas-liquid phase separation [20]. We confirm that this is the case also for our model: Figure 4(a) shows that the equation of state of the $A_{24}$ ring system in the fully-bonded limit is a monotonically-increasing function of the density, with the pressure increasing faster with density compared to an equivalent system made of chains. This is due to the rings exhibiting a stronger effective repulsion compared to chains, an effect that can be

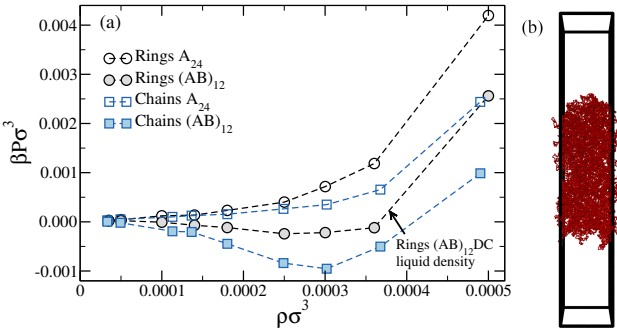

Figure 4: (a) A comparison between the equations of state of systems made of chains (blue squares) and rings (black circles) of type $A_{24}$ (empty symbols) and $(AB)_{12}$ (filled symbols). In all simulations we fix $\beta\epsilon = 20$. (b) A snapshot of a representative configuration of the direct-coexistence (DC) simulation of an $(AB)_{12}$ ring system. The average density of the liquid phase of the $(AB)_{12}$ ring system is indicated in panel (a) with an arrow.

traced back to the topological difference between the two architectures [33, 34].

Finally, we also consider rings decorated with reactive monomers of two (alternating) different types while keeping their number constant, obtaining an $(AB)_{12}$ ring system. Figure 4(a) shows that, in the fully-bonded limit ($\beta\epsilon = 20$), the $(AB)_{12}$ ring model exhibits an equation of state that is clearly non-monotonic. We again see the presence on an augmented repulsion with respect to the equivalent chain system, although the overall repulsion is not strong enough to suppress phase separation. Indeed, we confirm the presence of a gas-liquid phase separation by performing a direct-coexistence simulation, during the course of which we observe no melting of the interface (see Figure 4(b)). The enhanced formation of inter-molecular bonds thus favours the connectivity of the transient polymer network, demonstrating that the increased entropic cost of forming intra-molecular bonds in systems with multiple types of reactive monomers is sufficient to drive phase separation also in ring systems.

## 3 Conclusions

Polymer-based particles possess internal degrees of freedom that heavily contribute to the system's entropy and therefore to the determination of the bulk thermodynamic behaviour. If the system is composed of flexible, purely-repulsive objects, the entropic contribution dominates [35]. In the case of associative polymers, where some of the monomers are reactive and bond to each other, energy comes into play. However, in the fully-bonded limit, where the number of bonds is constant and independent of concentration, the entropic term becomes predominant again and fully controls the system thermodynamics. In this regime, the phase behaviour of associative polymers that form transient networks can be finely tuned by changing the type of reactive monomers, which in turn affects the ratio of intra- to inter-molecular bonds and therefore the overall inter-polymer large-scale connectivity. If the entropic gain of swapping intra-molecular with inter-molecular bonds is large enough, the system will phase separate [21]. Here we have shown numerically that this enhanced entropy binding is not a serendipitous effect, but it is a robust phenomenon with respect to thermal fluctuations, *i.e.* it is important even far from the fully-bonded regime, to disorder and to changes of the polymer architecture. Therefore, it can be leveraged to control the degree of association in synthetic systems, such as single-chain nanoparticle systems [16, 36, 37], or to shed light on the thermodynamics of systems that can be modelled, at least on a coarse-grained level, as

associative polymers [38–40]. As a further step forward, we plan to investigate how the size of the polymers and the number of reactive sites affect the material thermodynamics in future work. We believe our simulation results will stimulate the experimental verification, as well as the development of a theory able to reproduce this enhanced entropy binding effect.

## Acknowledgements

We thank Angel Moreno for fruitful discussions.

**Funding information** We acknowledge support from MIUR PRIN 2017 (Project 2017Z55KCW) and the CINECA award under the ISCRA initiative, for the availability of high performance computing resources and support (Iscra B "AssoPoN").

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
