# Peer review of "Entropy-driven phase behavior of associative polymer networks"

_SciPost Physics, doi:SciPost Phys. 15, 163 (2023)_

## Round 1 · Referee Report · Anonymous (Referee 2) · 2023-5-29

Strengths

The manuscript contributes to broadening our understanding of the phase behavior of an important class of materials.

Weaknesses

The analysis of the results could be improved

Report

The manuscript studies the phase behavior of polymers carrying reactive monomers that dimerize forming intra- and intermolecular, reversible bonds. Reactive chains are currently used to study multiple soft and biological systems including chromatin organization, phase separation of disordered proteins, or folding of single-chain polymeric nanoparticles.

The current submission is a follow-up of a previous contribution by the same group of authors (Phys. Rev. Lett. 2022 or Ref. 21). While increasing the chemical potential, systems of self-avoiding chains functionalized by a single type of (self-complementary) reactive monomers continuously transit from a diluted to a percolated phase without a discontinuity in density. In Ref. 21, the authors designed a system featuring a phase transition by decorating chains with two sets of self-complementary reactive monomers in which different types of monomers do not interact. Due to configurational terms, such a design penalizes intramolecular bonds in favor of intermolecular contacts. The current submission assesses the robustness of this finding by studying how changes in the original design impact the gas-liquid transition. In particular, chains with different topologies (ring vs linear) and organizations of the reactive monomers are considered.

The manuscript is certainly interesting in the way it contributes to broadening our understanding of the phase behavior of an important class of materials. The results are sound but it seems that their analysis could be improved as discussed below.

-The authors correlate the tendency of the system to phase separate with the fraction of bonds, pb. It seems that studying the fraction of intermolecular bonds would have been more insightful given that, because of intra-molecular bonds, pb seems to be maximized in the monofunctional system (which does not phase separate).

-It would be insightful to discuss how the critical point is affected by the number of reactive monomers and the chain’s length.

-It is not obvious to me that for disordered chains (low values of S) the configurational cost of forming loops is smaller; some loops will be smaller but others longer. Disordered chains may also find it harder to maximize the number of loops resulting in behaving as in the ‘defective’ arrangement of Fig. 3a.

-The ring topology has a major impact on the equation of state with the pressure of the A24 system at high density twice as big as the one in the linear topology. Beyond excluded volume considerations, I wonder if the ring topology also increases the propensity to form intramolecular bonds (e.g. with the appearance of zipped configurations).

-Following on from the previous point, the analysis of Fig. 3 and 4 could be improved by the study of the corresponding pb (ideally, disentangling intramolecular from intermolecular bonds). Similarly, the loop length distributions may help corroborate some of the claims made in the manuscript.

-I am wondering if the morphological and structural properties of the percolated phase (in particular the valency) are the same in all cases irrespective of the presence or not of the gas-liquid transition.

Minor

-Specify the protocol used to distribute the reactive monomers over the chain in the non-ordered system (multiple reactive monomers are allowed).

  • validity: good
  • significance: -
  • originality: -
  • clarity: -
  • formatting: -
  • grammar: -

Author:  Lorenzo Rovigatti  on 2023-08-08  [id 3882]

(in reply to Report 1 on 2023-05-29)
Category:
answer to question

Thanks for your remarks!
We have resubmitted a new version containing several changes as a result of your feedback. Here we attach the new figure 2, with an inset showing the "valency" of the polymers in the liquid state, as you suggested.
We have also added data and accompanying discussion on the distribution of intra-molecular loops.
We hope that our answers and the resulting changes (listed in the resubmission) will be satisfactory.

Attachment:

pd_BC24.pdf

---

## Round 1 · Referee Report · Anonymous (Referee 1) · 2023-7-21

Report

The manuscript addresses a very general bonding mechanism leading to transient polymer networks that occurs in diverse polymeric materials through a minimal modelling allowing the detection of the essential physical ingredients involved.

Within this framework, polymers capable of forming transient networks are flexible Kremer-Grest chains decorated with monomers able to form reversible bonds.
A previous paper from the same authors found that if monomers are of two different species arranged alternately on the chain, an entropy-driven first order phase transition appears between a regime dominated by intra-molecular towards one in which inter-molecular bonds are prevalent.
This finding is at odd with the case in which attractive monomers are all of the same species and the network formation takes place continuously.

The current paper reinforces and extends the findings of Ref.21 showing that previous results are robust against thermal fluctuations, against disorder, and changing the model topology, i.e. if rings are considered instead of chains. The paper is interesting for a large public , unraveling a fundamental mechanics, and the authors' conclusions are supported from the solid, rigorous and nicely organized results.
I therefore recommend its publication in SciPost Physics after addressing some minor comments I list hereafter.

-The authors employ chains of length $N_m=243$, differently from Ref.21 where they used $N_m=254$. This last accounts for 23 trunks made of 10 inert monomers and 24 active monomers. To me, it is not clear the length of the inert truck with the current choice.

-How does the phenomenology of Fig.1 compares quantitatively with the previous paper? At which $\beta\epsilon$ was previously investigated the system? A comment on this would help the comparison

-Please add legends (a,b) on figs.1,3. I believe legend of fig.3 incorrectly refers to the $S=1$ case for the inset

-As follows from the first point, how many inert monomers separate attractive monomers in the rings? Is this a constant number?

  • validity: -
  • significance: -
  • originality: -
  • clarity: -
  • formatting: -
  • grammar: -

Author:  Lorenzo Rovigatti  on 2023-08-08  [id 3883]

(in reply to Report 2 on 2023-07-21)
Category:
answer to question

Thank you for your comments! We have resubmitted a manuscript that contains several changes we made in response to your remarks. We hope that our answers and the resulting changes (listed in the resubmission) will be satisfactory.

---

## Round 2 · Author Response

Dear Editor,

Thank you for providing us with two constructive reports. We believe that taking into considerations the points raised by both Reviewers helped us to improve the clarity of the manuscript. The changes we carried out comprise adding a table and an inset to figure 2, as well as several new paragraphs of discussions.

We hope the attached reply and the changes reported therein will make the manuscript worth of publication in SciPost Physics.

Best regards,

Lorenzo Rovigatti and Francesco Sciortino

---

## Round 2 · List of Changes

Changes made in response to Reviewer's 1 remarks:

  • Added the following paragraph to discuss the new inset of figure 2:

"In order to gain more insight on the differences between the $A_{24}$ and $(AB)_{12}$ systems, we have also computed the number of inter-molecular bonds formed by each chain, $N_{\rm inter}$, for the same two systems of Figure~1. The inset of Figure~2(a) shows $N_{\rm inter}$ for the $A_{24}$ and $(AB)_{12}$ systems at $\rho\sigma^3 = 0.00049$ (the highest investigated density), as a function of the attraction strength $\beta epsilon$. At this high density no signs of phase separation have been detected in either systems. At the lowest value of $\beta \epsilon$ $N_{\rm inter}$ is comparable between the two systems, although in the $A_{24}$ more inter-molecular bonds are present in virtue of the higher bonding probability (see insets of Fig.~1). However, as the attraction reaches $\beta \epsilon = 10$, the number of inter-molecular bonds per chain in the $(AB)_{12}$ system, for which direct coexistence results demonstrate phase separation at lower density, experiences a steep increases that is not present in the $A_{24}$ system and that drives the transition. We also find that the $(AB)_{12}$ system with $\beta \epsilon = 10$ and $\rho\sigma^3 = 0.00025$, which, according to direct coexistence results, is near-critical and close to the density of the liquid phase, has a $N_{\rm inter} \approx 3$, making it a rather low-valence and sparsely-connected system."

  • added the following sentence to the conclusions:

"As a further step forward, we plan to investigate how the size of the polymers and the number of reactive sites affect the material thermodynamics in future work."

  • added the following paragraphs to discuss the new Table I:

"We start by considering the effect that the polymer architecture has on the average loop length, \textit{i.e.} the average chemical distance between two reactive monomers involved in an intra-molecular bond. Table I shows this quantity for ordered and (selected) disordered chains, as well as for rings, for the highest probed density ($\rho \sigma^3 = 0.00049$) and strongest attraction strength ($\beta \epsilon = 20$). As we will discuss and show below, under these conditions all the systems are homogeneous (no signs of phase separation) and essentially fully bonded, so that the average loop length can be considered a proxy for the entropic cost of forming an intra-molecular bond. Focussing on the chain systems (top four rows), it is clear that $(AB)$ chains tend to form larger loops compared to $A$ chains, which causes an increased polymer-polymer effective attraction that can drive phase separation [21]. However, in the presence of disordered arrangements of the reactive monomers the average loop length decreases. In the case of maximum disorder ($S = 1$), the average loop length of the $(AB)_{12}$ system is comparable to the value found for the ordered $A_{12}$ chains and, in fact, the consequential reduced cost of intra-molecular bonds is enough to suppress phase separation in the $(AB)_{12}$, $S = 1$ system (see below).

Table I also shows results for ring systems, for which going from one to two types of reactive monomers has the same qualitative effect observed for chains: the entropic cost of forming an intra-molecular bond goes up, and therefore the average loop length increases, causing a larger ring-ring effective attraction (see below for the consequences on the thermodynamics). However, the values in these case are larger than what observed in chains, owing to the smaller radius of gyration of (and therefore of the increased probability of intra-molecular contacts in) rings compared to chains of the same chemical size [32]."

  • added the following sentence to clarify how the non-ordered arrangements are randomly generated:

"In the case of disordered arrangements, the position of the reactive monomers are generated as follows: (i) a non-reactive monomer $i$ is selected at random; (ii) if $i$ is separated by more than $S$ inert monomers from any other reactive monomer then $i$ is flagged as a reactive monomer; (iii) if the number of reactive monomers is equal to the target $N_r$ value the procedure is stopped, otherwise steps (i)-(iii) are repeated until such a condition is fulfilled."

Changes made in response to Reviewer's 2 remarks:

  • added the following paragraph to the methods section:

"We study linear polymers (\textit{i.e.} chains) made of $N_m = 243$ monomers, with both ordered and disordered arrangements of the reactive monomers. The ordered chains start and end with $6$ inert monomers, and then the $N_r$ reactive monomers are placed equispatially, with $9$ inert monomers separating each pair of neighbouring reactive monomers. Note that we choose a slightly different system than the one in Ref.~21 to show that small differences in the number of inert monomers separating neighbouring reactive monomers (9 \textit{vs.} 10) and the presence of short inert segments at the beginning and at the end of the chain, absent in Ref.~21, do not alter the phenomenon we observe. In the case of disordered arrangements, the position of the reactive monomers are generated as follows: (i) a non-reactive monomer $i$ is selected at random; (ii) if $i$ is separated by more than $S$ inert monomers from any other reactive monomer then $i$ is flagged as a reactive monomer; (iii) if the number of reactive monomers is equal to the target $N_r$ value the procedure is stopped, otherwise steps (i)-(iii) are repeated until such a condition is fulfilled. By contrast, polymer rings are made by $N_m = 264$ monomers and $N_r = 24$ reactive monomers, with pairs of neighbouring reactive monomers being separated by $10$ inert monomers."

  • added the following two sentences to compare results discussed here with those in Ref. 21:

"In the following we will set $\beta \epsilon = 20$ as in Ref.~21 to obtain systems that are essentially fully bonded, or smaller values to probe thermal effects."

"Under these conditions, for which we find results compatible with those of Ref.~21, the effective attraction [...]"

---

## Editorial Decision

published